# Understanding the Immunopathology of HTLV-1-Associated Adult T-Cell Leukemia/Lymphoma: A Comprehensive Review

**DOI:** 10.3390/biom13101543

**Published:** 2023-10-19

**Authors:** Shingo Nakahata, Daniel Enriquez-Vera, M. Ishrat Jahan, Kenji Sugata, Yorifumi Satou

**Affiliations:** 1Division of HTLV-1/ATL Carcinogenesis and Therapeutics, Joint Research Center for Human Retrovirus Infection, Kagoshima University, Kagoshima 890-8544, Japan; 2Division of Genomics and Transcriptomics, Joint Research Center for Human Retrovirus Infection, Kumamoto University, Kumamoto 860-8556, Japan

**Keywords:** human T-cell leukemia virus type 1, adult T-cell leukemia/lymphoma, viral genes, genetic alterations, immune response, host–pathogen interaction, pathogenesis, treatment

## Abstract

Human T-cell leukemia virus type-1 (HTLV-1) causes adult T-cell leukemia/lymphoma (ATL). HTLV-1 carriers have a lifelong asymptomatic balance between infected cells and host antiviral immunity; however, 5–10% of carriers lose this balance and develop ATL. Coinfection with *Strongyloides* promotes ATL development, suggesting that the immunological status of infected individuals is a determinant of HTLV-1 pathogenicity. As CD4+ T cells play a central role in host immunity, the deregulation of their function and differentiation via HTLV-1 promotes the immune evasion of infected T cells. During ATL development, the accumulation of genetic and epigenetic alterations in key host immunity-related genes further disturbs the immunological balance. Various approaches are available for treating these abnormalities; however, hematopoietic stem cell transplantation is currently the only treatment with the potential to cure ATL. The patient’s immune state may contribute to the treatment outcome. Additionally, the activity of the anti-CC chemokine receptor 4 antibody, mogamulizumab, depends on immune function, including antibody-dependent cytotoxicity. In this comprehensive review, we summarize the immunopathogenesis of HTLV-1 infection in ATL and discuss the clinical findings that should be considered when developing treatment strategies for ATL.

## 1. Introduction

Human T-cell leukemia virus type 1 (HTLV-1) is a retrovirus that causes various diseases, including adult T-cell leukemia/lymphoma (ATL), HTLV-1–associated myelopathy/tropical spastic paraparesis (HAM/TSP), and HTLV-1 uveitis (HU). HTLV-1 mainly infects CD4+ T cells and can transform the infected cells to cause ATL, which has a poor prognosis [1,2]. ATL is classified into indolent (smoldering and chronic) and aggressive types (acute and lymphoma). Patients with aggressive ATL have a median survival time of less than one year, and those with indolent ATL often undergo a blast crisis after several years [3,4,5,6,7]. Worldwide, approximately 5–10 million individuals are infected with HTLV-1 (carriers), geographically concentrated in Japan, South America, Africa, and the Caribbean. The age of ATL onset varies regionally, from an average of 47–49 years in South America to 68 years in Japan, with a slightly higher prevalence in men than in women [8,9,10]. The main routes of HTLV-1 infection differ by country. In Japan, mother-to-child transmission via breast milk predominates, whereas in South America and Africa, horizontal transmission is also common [11,12,13]. ATL cells have a mature T-cell phenotype with characteristic multi-lobulated nuclei, known as “flower cells,” and the typical activated T-cell phenotype of ATL cells includes the surface markers CD4+ and CD25+ [14].

The genomic structures of HTLV-1 and the human immunodeficiency virus (HIV) are similar, with structural genes containing Gag, Pol, and Env; regulatory genes, such as HIV-1 transactivator of transcription (Tat) and Rev of HIV; Tax and Rex of HTLV-1; accessory genes such as Vif, Vpr, Vpx, Vpu, and Nef of HIV; p12, p13, p30, and the HTLV-1 bZIP factor (HBZ) of HTLV-1, which are flanked by 5′ and 3′ long terminal repeats (LTR) [15]. Pathogenicity differs depending on the genotype in these accessory and regulatory regions. HIV mainly infects the CD4 receptor, whereas HTLV interacts with glucose transporter 1, neuropilin-1, and heparan sulfate proteoglycans to enter target cells. In addition to CD4+ T cells, both HIV and HTLV-1 infect non-lymphoid monocytic cells, such as macrophages and dendritic cells, and cause diseases originating from CD4+ helper T-cells [16]. During HIV infection, direct cell destruction because of HIV infection and the proliferation and destruction of infected cells by HIV antigen-specific CD8+ cytotoxic T-cells (CTLs) decreases the CD4+ T-cell count and consequently causes acquired immunodeficiency syndrome (AIDS) [17]. In contrast, HTLV-1 remains dormant in the host genome as a provirus for several decades, resulting in a persistent infection in which 1–2% of peripheral blood mononuclear cells are HTLV-1-infected lymphocytes [18,19]. These infected lymphocytes become cancerous in some HTLV-1 carriers, causing ATL. Therefore, the onset of ATL cannot be explained by viral infection alone, and host factors are thought to be involved in this process.

HTLV-1 encodes the viral transcription transactivator, Tax, in the pX region of its genome, which promotes oncogenesis. Tax interacts with various host cell proteins, affects intracellular signaling pathways, regulates gene transcription, and contributes to HTLV-1-infected T-cell proliferation [20,21]. However, because Tax exhibits strong immunogenicity, infected T cells expressing Tax are actively eliminated by Tax-specific CTLs [22,23]. At the onset of ATL, approximately 50% of cases do not show Tax expression due to methylation or the deletion of the 5′ LTR region; thus, the selective proliferation of clones that are not attacked by CTLs is likely to be promoted [24]. In addition, recent studies have revealed that sense-side transcription leading to Tax expression occurs in transient bursts, even in the absence of genomic deletions or epigenetic changes in the provirus [25,26,27]. This may be a viral survival strategy that minimizes Tax expression to avoid a CTL attack. HTLV-1 encodes the oncogenic factor HBZ via antisense transcription of the pX region. HBZ promotes the transcription of forkhead box protein P3 (Foxp3), the master regulator of regulatory T cells, and confers regulatory T-cell traits to HTLV-1-infected T cells [28]. As a result, the proliferation of HTLV-1-infected T cells promotes immunosuppression via the secretion of inhibitory cytokines, such as interleukin (IL)-10. Changes in the innate immune system, such as macrophages and eosinophils, have been observed in patients with ATL [29,30]. Therefore, host-specific immunity may be involved in ATL development. Age, HTLV-1 proviral load (PVL), human leukocyte antigen (HLA) haplotype, and strongyloidiasis are known risk factors for ATL development [19,31,32,33]. This suggests a relationship between immunity and carcinogenesis.

Clinically, patients with ATL develop profound immunodeficiency, similar to that seen in patients with AIDS [34]. Furthermore, in patients with acute and lymphoma types, antibiotics are generally administered to prevent infections because the immune function is weakened by treatments such as intense chemotherapy and transplantation. In patients with ATL, complications such as hypercalcemia and opportunistic infections are associated with a poor prognosis [35]. Opportunistic infections may include *Pneumocystis carinii*, aspergillosis, and candidiasis, which are observed in patients with AIDS, and cytomegalovirus pneumonia and *Strongyloides stercoralis* infection [36,37]. Opportunistic malignancies, including Kaposi’s sarcoma and Epstein–Barr virus-associated B-cell lymphoma, which are common in patients with AIDS, have also been reported in patients with ATL [38,39]. Importantly, opportunistic infections have also been reported in HTLV-1 carriers, and the carriers of opportunistic infections may be more susceptible to ATL [40,41]. Thus, immunity is closely related to disease progression in HTLV-1 carriers and prognosis in patients with ATL. Therefore, the elucidation of the immunopathogenesis could enhance the understanding of ATL onset mechanisms and the establishment of therapeutic methods for its treatment. However, few studies have comprehensively integrated and discussed the knowledge accumulated from basic and clinical research. In this review, we focus on immune abnormalities in the HTLV-1 infection and ATL onset, summarize the findings reported to date from both basic research and clinical studies, and we discuss them in an integrated manner from the perspective of ATL therapeutics.

## 2. Immune Disruption Because of HTLV-1 Infection

Approximately 95% of HTLV-1 carriers remain asymptomatic throughout their lives, without any clinical symptoms associated with HTLV-1 infection. Therefore, unlike HIV, which continues to proliferate actively even during the asymptomatic period, HTLV-1 is a latent infection, which suggests that the HTLV-1 infection alone does not affect the infected host. However, several studies have shown that HTLV-1 can evade and disturb the host immune system. These processes are described below.

### 2.1. T-Cell Anergy Induced by HTLV-1 Infection

Sense-strand transcription of HTLV-1 is generally silent; however, it is activated in ex vivo cultures under hypoxic stress or high-glucose conditions [42,43]. The plus strand appears to be silent in the peripheral blood but is expressed in the lymph nodes, leading to de novo cell-to-cell infection [43]. Zinc-finger CCCH-type antiviral protein 1 (ZC3HAV1/ZAP) is a host antiviral factor that suppresses translation and promotes the degradation of specific viral mRNAs by binding to cytosine-guanine (CG)-rich RNA sequences. HTLV-1 sequences are rich in CG dinucleotides and are targets of ZAP [44], which may be a mechanism underlying the repression of viral transcription.

HTLV-1 encodes a potent transcription factor, Tax, which strongly induces viral gene expression by enhancing the recruitment of enhancers such as p300 to the viral LTR [1]. Tax also transforms T cells and fibroblasts in vitro, indicating that it functions as an oncogene [45,46]. Tax interacts with intracellular signaling pathways and transcription factors in infected T cells to regulate the cell cycle, cell proliferation, and apoptosis. Tax induces the expression of genes encoding various cytokines and chemokines. In in vitro experiments, Tax expression promoted the production of inflammatory cytokines, such as IL-2, IL-6, and tumor necrosis factor (TNF)-α, and the immunosuppressive cytokine IL-10 [47,48]. Furthermore, Tax ectopically upregulates HLA class II expression in HTLV-1-infected T cells [49]. CD4+ T cells infected with HTLV-1 are in a state of excessive activation and express HLA class II molecules, which are normally expressed by only a small fraction of T cells, and may present antigens as antigen-presenting cells (APCs). HTLV-1-infected T cells may become tolerogenic APCs and cause antigen-specific T-cell tolerance (anergy) [49].

### 2.2. Acquisition of Regulatory T-Cell Phenotype by HTLV-1 Infection

HBZ has low immunogenicity and is persistently expressed during the viral incubation period and ATL onset and progression [50,51]. HBZ is a viral factor essential for ATL oncogenesis and functions at the protein and RNA levels [52]. HBZ induces the expression of Foxp3 and regulatory T-cell-related molecules, such as CC chemokine receptor 4 (CCR4) and T-cell immunoreceptors with Ig and immunoreceptor tyrosine-based inhibitory motif (ITIM) domains (TIGIT) [53]. HBZ also induces inflammatory pathology in addition to T-cell tumorigenesis in *HBZ* transgenic mice [54]. Furthermore, HBZ in HTLV-1 infected T cells suppresses the expression or function of the inhibitory co-receptors TIGIT, programmed cell death 1 (PD-1), B and T lymphocyte attenuator (BTLA), and leukocyte-associated Ig-like receptor 1, thereby enhancing T-cell receptor signaling and promoting the proliferation of infected cells [55].

Additionally, Rex, p30, p12, and p13 are transcribed from the sense strand of the pX region HTLV-1 via alternative RNA splicing. The CTLs recognize HLA class I presented with the Tax peptide, and p12 expression reduces the plasma membrane expression of HLA class I [56], likely promoting CTL evasion. In contrast, p30 expression decreases the expression of PU.1, a factor essential for the development of granulocytes, monocytes, macrophages, and lymphoid cells [57]. PU.1 downregulation reduces the Toll-like receptor 4 (TLR4) expression and suppresses the secretion of pro-inflammatory cytokines, including monocyte chemoattractant protein-1, TNF-α, and IL-8 in macrophages [57]. Thus, multiple mechanisms involving viral factors suppress the immune response caused by HTLV-1 infection, and the infected cells continue to survive without being eliminated (Figure 1).

## 3. Immune Abnormalities in ATL

### 3.1. Genetic Alterations

A large-scale integrative genomic analysis of ATL revealed that approximately 90% of ATL cases were enriched with activating mutations in the T-cell receptor (TCR)–nuclear factor (NF)-κB pathway [58]. Furthermore, using comprehensive DNA methylation analysis, the hypermethylation of CpG islands, also known as CpG island methylator phenotype hypermethylation (CIMP), has been observed in approximately 40% of ATL cases [58]. In addition, genetic abnormalities, such as mutations and deletions, accumulate in HLA class I genes [58]. Genomic aberrations targeting the 3′-untranslated region (UTR) of programmed death ligand 1 (PD-L1) have also been reported to accumulate in ATL [59]. Truncation of the 3′-UTR of the PD-L1 gene occurs as a result of structural abnormalities, such as the deletion, translocation, and inversion of the PD-L1 3′-UTR genomic region, resulting in increased PD-L1 mRNA expression [59]. These findings lead to the speculation that during the carcinogenic process of ATL, clones accumulate genomic abnormalities that can evade tumor immunity, such as a selective increase in the number of HLA class I and PD-L1 aberrations. ATL cases with CIMP or PD-L1 amplification have a high degree of malignant phenotypes [58,60].

In addition, single-cell analysis revealed that HTLV-1-infected cells that acquired clonal proliferation ability exhibited the abnormal expression of various immune-related molecules [61]. In addition to the upregulation of the immunosuppressive molecules PD-L1, CD73, and CD39, the upregulation of activation markers, such as CD71, CD25, and CD38, and co-stimulatory molecules, such as CD99, CD28, and CD278, was observed [61]. In HTLV-1-infected T cells, before clonal expansion, the expression of HLA class II molecules increased, but their expression decreased during clonal expansion [61]. Therefore, ATL cells appear to evade antitumor immunity by expressing immunosuppressive molecules with activated phenotypes (Figure 1).

### 3.2. Tumor Immune Microenvironment

Tumor-associated macrophages (TAMs) are major constituents of tumor stromal cells and act as tumor promoters by promoting cancer cell proliferation, invasion, angiogenesis, and immunosuppression [62,63,64]. TAM infiltration is frequently present in acute and lymphoma-type ATL and is associated with poor prognosis [65]. Phagocytosis by TAMs is suppressed when the cells express CD47, a ligand for SIRPα of TAMs, which is known as a “don’t eat me” signal [66]. Although CD47 is abundantly expressed in erythrocytes and platelets, tumor cells also express CD47 to prevent phagocytosis by macrophages [66]. The expression of CD47 on ATL cells has not been shown to affect prognosis, whereas the expression of SIRPα on stromal cells is paradoxically associated with a favorable prognosis [67].

PD-L1 is a transmembrane protein that suppresses immune responses by binding to two inhibitory receptors, PD-1 and B7-1 (CD80), and it is strongly expressed on the surface of tumor and non-transformed cells in the tumor microenvironment [68]. PD-L1 expression in ATL cells has also been observed in ATL lymph node lesions and is a poor prognostic factor [69]. PD-L1 expression in stromal cells, including macrophages and dendritic cells, is associated with a favorable prognosis in patients with ATL [69]. However, the biological role of PD-L1 expression in the tumor microenvironment remains unclear. Additionally, the proportions of invariant natural killer T, natural killer (NK), and dendritic cells are decreased in the peripheral blood of patients with ATL [70].

Single-cell analyses have shown that ATL is accompanied by a decrease in B cells and an increase in myeloid cells [61]. Among the myeloid cells, dendritic cells and atypical monocytes are upregulated in patients with ATL. These myeloid cells are also associated with increased activation markers, such as CD64, and immune checkpoint molecules, such as PD-1 [61]. Although the B-cell count tends to decrease in patients with ATL, the interferon pathway is upregulated in myeloid cells [61]. Importantly, CTL decline is associated with genetic aberrations in PD-L1, and in such cases, PD-L1 is upregulated in B cells and myeloid cells in the immune microenvironment [61]. Furthermore, NK cells exhibit functional abnormalities in HTLV-1 carriers and patients with ATL [61]. These findings suggest that ATL carcinogenesis is accompanied by changes in the immune microenvironment [61] (Figure 1).

## 4. Clinical Aspects of HTLV-1 Infection and ATL Related to Immunological Alterations

### 4.1. Mother-to-Child Transmission

ATL occurs in less than 5% of HTLV-1 carriers after five to six decades of chronic infection. Moreover, HTLV-1 infection during childhood, specifically more than 6 months of breastfeeding (relative risk, 2.91; 95% confidence interval, 1.69–5.03) is a risk factor for ATL transformation [71]. This type of transmission has been associated with increased maternal PVL after delivery [72] and a genetic predisposition to a major locus on chromosome 6q27 in children [73]. Moreover, an increase in APCs in breast milk may be responsible for HTLV-1 infection during extended breastfeeding [74]. In addition, pregnant women coinfected with *Strongyloides stercoralis* have a high risk of transmission [75].

### 4.2. Coinfection

#### 4.2.1. HIV

HIV and HTLV belong to the Retroviridae family and share genomic and epidemiological characteristics. Moreover, a recent meta-analysis showed that individuals with HIV and HTLV-I coinfection have accelerated HIV progression, worse survival, and a high incidence of HAM/TSP, peripheral neuropathy, encephalopathy, and opportunistic infections, such as scabies, candidiasis, and strongyloidiasis [76,77]. HTLV-1 activates CD4+ T cells, and Tax upregulates HIV-1 infection [78]. In contrast, HIV coinfection with HTLV-2, another type of HTLV that is non-pathogenic but rarely associated with HAM/TSP, is associated with slow T-cell depletion and AIDS progression, with higher HIV-1 viral control and lower mortality rates [79]. This effect is mediated by CTL expansion induced by HTLV-2, including a high rate of the effector memory phenotype and increased levels of granzymes A and B and perforin [79]. Although a case of ATL has been reported in an HIV-1-positive patient [80], more evidence is needed to determine whether individuals with HIV infection have an increased risk of ATL.

#### 4.2.2. Strongyloides

*S. stercoralis* is an opportunistic nematode that can cause severely disseminated infections in immunocompromised individuals. It is typically found in tropical regions, and its area of distribution overlaps with that of HTLV-1 [81]. HTLV-1 infections have been linked to severe strongyloidiasis, owing to a high production of interferon (IFN)-γ and the downregulation of IL-4, IL-5, IL-13, and immunoglobulin E (IgE), which are necessary for host defense against parasites [82]. Additionally, the HTLV-1 PVL is significantly higher in HTLV-1 carriers with strongyloidiasis than in those without strongyloidiasis [41]. Strongyloidiasis can induce the polyclonal expansion of HTLV-1-infected T cells via the upregulation of the IL-2/IL-2R pathway [83]. *S. stercoralis* infections may partially explain the clinical and genomic disparities in ATL cases between Japan and other regions. In countries other than Japan, *S. stercoralis* infections are associated with early-onset ATL [84].

### 4.3. Prognosis

Owing to the resistance to chemotherapy, high relapse rates, and opportunistic infections, ATL prognosis remains poor even with aggressive treatment, with a 3-year overall survival of less than 25%. Multiagent chemotherapy, interferon-α plus zidovudine, mogamulizumab, and allogeneic hematopoietic stem cell transplantation (allo-HSCT) remain the main treatment modalities. Mogamulizumab is a monoclonal antibody that targets CCR4 and is associated with a higher response rate in combination with chemotherapy. It may transiently improve the response before allo-HSCT, which is the only curative treatment available for patients with aggressive ATL. However, not all patients are candidates for allo-HSCT. In well-conducted clinical trials, only 7% of patients with aggressive ATL undergo allo-HSCT [85]. The 5-year survival is 40% in patients who undergo allo-HSCT compared with 12% in patients who do not receive allo-HSCT [85].

### 4.4. Prognostic Factors in Aggressive ATL Subtypes

Several scoring methods have been developed worldwide using host-dependent and disease-burden factors. The ATL Prognostic Index was constructed retrospectively for patients with aggressive subtypes who did not undergo allo-HSCT [86]. Adverse host prognostic factors include age (>70 years), serum albumin (<3.5 g/dL), and poor performance status (Eastern Cooperative Oncology Group [ECOG] score >1); disease-burden factors include Ann Arbor stage (I–II vs. III–IV) and soluble interleukin-2 receptor (sIL-2R) >20,000 U/mL. The Japan Clinical Oncology Group also developed a prognostic index that included only the performance status and corrected calcium levels (>2.75 mmol/L) based on a combined analysis of three consecutive clinical trials. However, the patients in these trials were relatively young, physically fit, and without organ dysfunction [85]. These prognostic indices indicate a median overall survival of less than 8 months in patients at high risk.

A nationwide survey in Japan evaluated patients with aggressive ATL undergoing intensive treatment with potential indications for allo-HSCT. The independent prognostic factors identified were clinical subtype (acute, worse than lymphoma), poor performance status (ECOG score > 1), corrected calcium levels (≥12 mg/dL), C-reactive protein level ≥ 2.5 mg/dL, and sIL-2R level > 5000 U/mL [87]. The stem cell source and disease status during transplantation are associated with better outcomes in allo-HSCT [88].

### 4.5. Anti-HTLV-1 Immunity

#### 4.5.1. Tax-Specific CTLs

Tumor-specific CTL induction is associated with the prognosis of solid tumors [89]. In patients with ATL, Tax-specific CTLs increase after allo-HSCT and mogamulizumab treatment [90,91]. Tax-CTLs are found in long-term allo-HSCT survivors. Post-transplant induction and increased Tax-CTLs may contribute to the graft-versus-leukemia effect in long-term survivors [91]. Skin rash induced by mogamulizumab is associated with better survival. This may be due to CD8+ T-cell infiltration in skin lesions and a Tax-specific CTL increase [90]. A reversal of the CD4-to-CD8 ratio in peripheral mononuclear cells has been reported in long-term survivors [90]. Moreover, Tax-CTLs occur more frequently after severe herpes simplex virus infections during or after chemotherapy or mogamulizumab treatment [90]. A recent study showed preliminary positive results following varicella-zoster virus vaccination, including an increase in Tax-CTLs after mogamulizumab-based treatment in patients with aggressive ATL subtypes [92]. Thus, the herpes simplex virus and the varicella-zoster virus may induce CTL activation via cytokine and chemokine release. However, the exhaustion of Tax-CTLs is correlated with the HTLV-1+ T-cell count and CTL activity [93]. These findings suggest that functional Tax-CTLs are important prognostic factors for the treatment of patients with ATL. Therefore, considering that the CTL decrease in patients with ATL is associated with genetic aberrations of PD-L1 in ATL cells [61], interactions between the immune microenvironment and ATL cells may be important to CTL function, and thus, for post-transplant outcomes (Figure 2).

#### 4.5.2. HBZ-Specific CTLs

The graft-versus-leukemia effect of allo-HSCT has been explained primarily by Tax-specific CTLs. Although Tax is expressed in only approximately 50% of patients with ATL, HBZ transcripts are present in all ATL cases. HBZ-specific CTLs have also been induced in vitro without showing limited cytotoxic activity against ATL cells [94]. Additionally, Tax peptides have stronger binding characteristics than HBZ peptides. Consequently, HTLV-1-infected individuals have a higher frequency of Tax-specific CTLs [94]. However, HLA class I alleles related to high-affinity HBZ peptides may play a protective role in asymptomatic HTLV-1 carriers and patients with HAM/TSP [95]. This effect has been explained by the correlation between a low PVL and the binding affinity of HBZ peptides for HLA class I [95]. Moreover, a study reported that HBZ could induce CD4+ T-cell responses in some patients with ATL, but only after allo-HSCT [96]. The HBZ peptide RRRAEKKAADVA (HBZ_114–125_) sequence with HLA-DRB1*15:02 restriction is recognized by specific CD4+ T cells [96]. Further studies are required to determine the immunogenicity and protective efficacy of HBZ peptides against ATL.

## 5. Development of Therapeutic Drugs Targeting Immunity for ATL

### 5.1. Anti-CCR4 Antibody Therapy (Mogamulizumab)

Mogamulizumab, an anti-CCR4 antibody, was the first molecular-targeted drug approved for ATL treatment in Japan. Its effects include antibody-dependent cellular cytotoxicity and the enhancement of antitumor immunity via regulatory T-cell suppression [97,98]. Mogamulizumab has been favored in clinical practice because of its ability to control disease activity in patients with aggressive ATL; however, consensus is lacking as to whether it provides additional benefits when administered with chemotherapy [99,100,101]. In addition, in patients with ATL treated with mogamulizumab, death from acute graft-versus-host disease after transplantation is often associated with regulatory T-cell suppression, and mogamulizumab should be used with caution in transplant-eligible patients [102,103] (Figure 2). Mogamulizumab may be effective when combined with other treatments.

### 5.2. Anti-PD-1 Antibody Therapy (Nivolumab)

The anti-PD-1 antibody exerts an antitumor effect via binding to PD-1 on T cells and inhibits the transduction of inhibitory signals via inhibiting PD-1 binding to PD-L1/PD-L2, resulting in T-cell activation. It is effective in treating solid tumors such as malignant melanoma and lung cancer [104]. Although clinical trials using anti-PD-1 antibody have been conducted in patients with ATL, the effectiveness of nivolumab for treating ATL has not been confirmed. Conversely, the administration of nivolumab can exacerbate disease conditions [105]. Anti-PD-1 antibody treatment has been reported to convey ATL cells with properties similar to those of tumor-infiltrating regulatory T cells [106]. As various abnormalities occur in the immune environment of patients with ATL [61], the benefit of anti-PD-1 antibody administration may not be obtained in some patients with ATL, and further research is required to assess whether anti-PD-1 antibody therapy has clinical application.

### 5.3. Lenalidomide

Lenalidomide, a thalidomide derivative, is an immunomodulatory agent used in the treatment of multiple myeloma. Lenalidomide has been reported to have immunopotentiating activity against multiple myeloma, including a direct tumor cell-killing effect and enhanced T and NK cell function [107]. In ATL, the inhibition of the enhancer of zeste homolog 2 (EZH2) expression induces the phosphorylation of the signal transducer and activator of transcription 3, resulting in cell growth suppression [108]. A phase II trial of lenalidomide in patients with recurrent or relapsing ATL showed clinically significant antitumor activity and acceptable toxicity [109]. The effect of lenalidomide on ATL requires further study. The elucidation of its mechanism of action in ATL would enable the development of more effective lenalidomide therapy.

### 5.4. Dendritic-Cell-Based Vaccine Therapy

Dendritic-cell vaccine therapy efficiently induces tumor antigen-specific CTLs and administering tumor antigen peptide-pulsed dendritic cells exerts an antitumor effect in patients with cancer [110]. A dendritic-cell vaccine targeting Tax has been developed for ATL [111]. The vaccine is prepared via culturing and differentiating monocytes obtained from the patient blood into dendritic cells, followed by the addition of Tax antigen peptides to the dendritic cells [111]. As dendritic-cell vaccine preparation is autologous, it can be expected to effectively activate CTLs against ATL cells. A phase I trial has reported good safety and efficacy [111], and this cancer immunotherapy has potential as a future treatment for ATL. As in the case of allo-HSCT, the therapeutic effect of dendritic cell therapy depends on CTL function; therefore, establishing a therapeutic method is necessary, taking into account the genetic abnormalities, such as PD-L1 in ATL cells (Figure 2).

## 6. Conclusions

Antiviral therapy, multidrug chemotherapy, and allo-HSCT are the main forms of therapy for ATL. However, regardless of the treatment method used, controlling patient conditions is challenging, and further improvement is required. Among the therapeutic drugs for ATL, the effects of mogamulizumab, lenalidomide, and allo-HSCT are strongly dependent on the patient’s immune function (Figure 2). Recent basic research has revealed that the immune function of patients with ATL is weakened. Further elucidation of the pathology is expected to lead to the establishment of a curative treatment for ATL. Chimeric antigen receptor T-cell therapy is also being developed for various types of leukemia; however, a scientific understanding of the patient’s immune status and correlating it with ATL treatment is important.

## Figures and Tables

**Figure 1 biomolecules-13-01543-f001:**
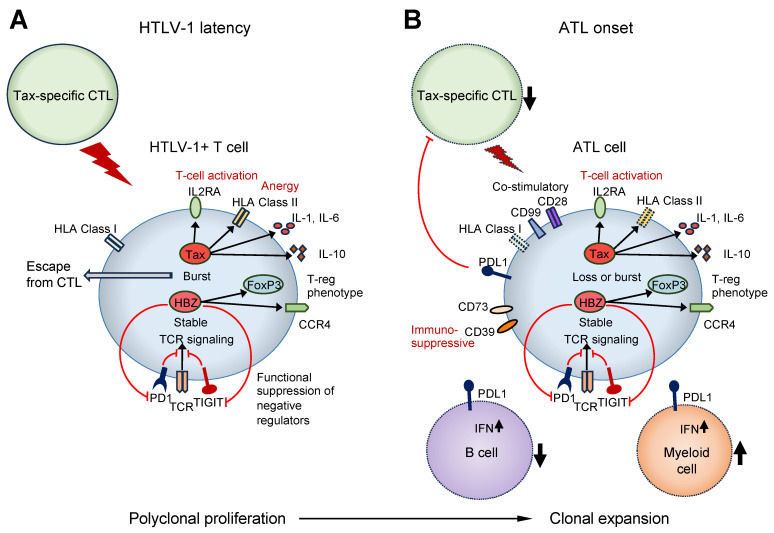
Summary of the immune pathologies during HTLV-1 latency and onset of ATL. (**A**) In asymptomatic HTLV-1 carriers, the proliferation of HTLV-1-infected T cells and their clearance by CTLs are in equilibrium. HTLV-1 transiently expresses Tax and affects the expression and function of immune-related molecules to evade host immunity and promote survival of HTLV-1-infected cells. (**B**) Patients with ATL develop immunodeficiency; however, ATL cells also accumulate genetic abnormalities in immune-related genes, and clones that are advantageous for escaping host immunity are selected for extensive proliferation. In addition to CD4+ T cells, decreased numbers of CTLs and B cells, increased numbers of myeloid cells, and decreased NK cell function are observed. HTLV-1, Human T-cell leukemia virus type 1; ATL, adult T-cell leukemia/lymphoma; CTL, cytotoxic T cell.

**Figure 2 biomolecules-13-01543-f002:**
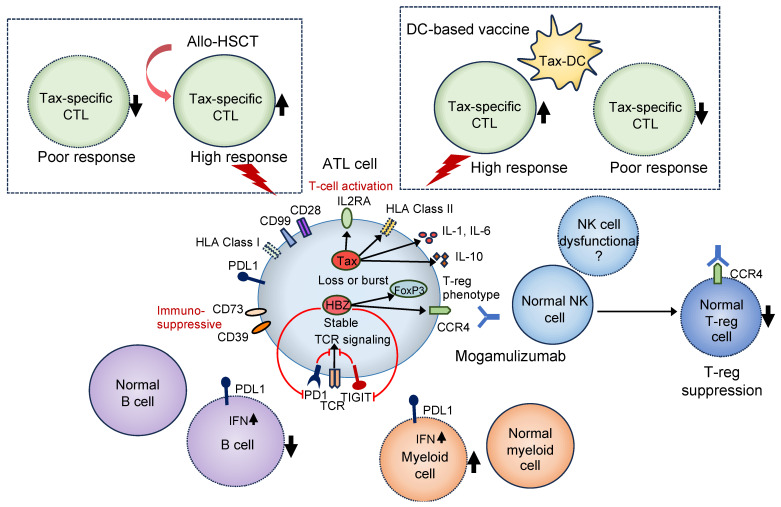
Relationship between immune pathology and therapeutic efficacy in patients with ATL. As the therapeutic effects of allo-HSCT and Tax-dendritic cell vaccine therapy depend on the activity of Tax-specific CTLs, the presence of functional CTLs is indispensable. Characterization of circulating ATL cells may be important as CTL decline is also associated with genomic abnormalities in ATL cells, such as PD-L1 [61]. Mogamulizumab suppresses not only ATL cells but also regulatory T cells (Treg). The antibody-dependent cellular cytotoxic effect of mogamulizumab depends on the activity of NK cells and may be less effective in some patients with ATL, owing to decreased NK cell function [61,70]. Dotted lines in boxes indicate the hypotheses. ATL, adult T-cell leukemia/lymphoma; allo-HSCT, allogeneic hematopoietic stem cell transplantation; PD-L1, programmed death ligand 1; CTL, cytotoxic T cell; NK, natural killer cell; DC, dendritic cell.

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
