# Peer review of "Understanding the Immunopathology of HTLV-1-Associated Adult T-Cell Leukemia/Lymphoma: A Comprehensive Review"

_biomolecules, 2023, doi:10.3390/biom13101543_

Round 1
Reviewer 1 Report
The review by Nakahata et . summarizes the contributions of recent scientific research in the study of immunologic response in the context of HTLV-infected individuals who will develop ATL and therapeutical approaches. Data comparing the literature are well represented, controversial findings are discussed, and limitations and perspectives of studies in this field are described. Adequate and detailed are the figures. The review is of great interest in the field of ATL study.
I could make two small revision suggestions:
Insert some references on published works, including reviews on Tax molecular interactions and effects in cellular pathways (e.g. NF-kB).
It might be helpful to indicate in the caption of Figure 2 what DC is.
Author Response
The review by Nakahata et . summarizes the contributions of recent scientific research in the study of immunologic response in the context of HTLV-infected individuals who will develop ATL and therapeutical approaches. Data comparing the literature are well represented, controversial findings are discussed, and limitations and perspectives of studies in this field are described. Adequate and detailed are the figures. The review is of great interest in the field of ATL study.
I could make two small revision suggestions:
Insert some references on published works, including reviews on Tax molecular interactions and effects in cellular pathways (e.g. NF-kB).
Our reply)
We appreciate for valuable comments and suggestions.
In response to the reviewer’s comment, we cited the following references in the Introduction section (Page 2, line 68 and Page 11, lines 462 to 464).
Mohanty, S.; Harhaj, E.W. Mechanisms of Oncogenesis by HTLV-1 Tax. Pathogens 2020, 9, 543.
Currer, R.; Van Duyne, R.; Jaworski, E.; Guendel, I.; Sampey, G.; Das, R.; Narayanan, A.; Kashanchi, F. HTLV tax: a fascinating multifunctional co-regulator of viral and cellular pathways. Front Microbiol. 2012, 3, 406.
In addition, previous reference 34 below was removed from the reference list because it was not cited in the text.
Gonçalves, D.U.; Proietti, F.A.; Ribas, J.G.; Araújo, M.G.; Pinheiro, S.R.; Guedes, A.C.; Carneiro-Proietti, A.B. Epidemiology, treatment, and prevention of human T-cell leukemia virus type 1-associated diseases. Clin Microbiol Rev. 2010, 23, 577-589.
It might be helpful to indicate in the caption of Figure 2 what DC is.
In response to the reviewer’s comment, we added the caption “DC, dendritic cell” in the figure legend (Page 8, line 328).
Reviewer 2 Report
This review article covers how HTLV-1 produces adult T-cell leukemia/lymphoma. Although 5-10% of HTLV-1 carriers develop ATL, they maintain a lifelong asymptomatic balance between infected cells and host antiviral immunity. HTLV-1 dysegulation of CD4+ T cells, which are essential for host immunity, increases infected T cell immunological evasion. Throughout ATL development, host immunity-related gene genetic and epigenetic modifications progressively impair immunological conditions. Various therapy are available to address these concerns. The sole treatment for ATL is allogeneic hematopoietic stem cell transplantation, however the patient's immune system may interfere with it. Mogamulizumab, an anti-CC chemokine receptor 4 antibody, is similarly dependent on immunological activities such as antibody-dependent cytotoxicity.
This excellent review outlines the immunological origin of HTLV-1 infection in ATL and incorporates clinical findings to predict ATL treatment considerations.
Not applicable
Author Response
As instructed, the whole manuscript has been proofread and checked that it complies with the requirements for submission to Biomolecules.